RESEARCH

# Adequate statistical power in clinical trials is associated with the combination of a male first author and a female last author

**Abstract** Clinical trials have a vital role in ensuring the safety and efficacy of new treatments and interventions in medicine. A key characteristic of a clinical trial is its statistical power. Here we investigate whether the statistical power of a trial is related to the gender of first and last authors on the paper reporting the results of the trial. Based on an analysis of 31,873 clinical trials published between 1974 and 2017, we find that adequate statistical power was most often present in clinical trials with a male first author and a female last author (20.6%, 95% confidence interval 19.4-21.8%), and that this figure was significantly higher than the percentage for other gender combinations (12.5-13.5%; P<0.0001). The absolute number of female authors in clinical trials gradually increased over time, with the percentage of female last authors rising from 20.7% (1975-85) to 28.5% (after 2005). Our results demonstrate the importance of gender diversity in research collaborations and emphasize the need to increase the number of women in senior positions in medicine.
DOI: https://doi.org/10.7554/eLife.34412.001

**WILLEM M  OTTE, JOERI K  TIJDINK, PAUL L  WEERHEIM, HERM J  LAMBERINK AND CHRISTIAAN H  VINKERS***

**\*For correspondence:** c.h. vinkers@umcutrecht.nl

**Competing interests:** The authors declare that no competing interests exist.

## Introduction

There is increasing awareness that many clinical trials have systematic methodological flaws and that their results may be biased, exaggerated, and difficult to reproduce (*Ioannidis et al., 2014*). Clinical trials are the result of complex group efforts. Male and female researchers differ in their collaborative strategies which depends on the level of their expertise and whether they have a junior or senior position (*Zeng et al., 2016*; *Bozeman and Gaughan, 2011*). There are indications that mixed gender teams may make the best use of personal knowledge and skills, (*Nielsen et al., 2017*) an effect also reported in a scientific research context (*Woolley et al., 2010*; *Campbell et al., 2013*). Even though this may in turn positively influence the quality of clinical research, (*Nielsen et al., 2017*) no studies have systematically investigated whether collaborations between male and female researchers affect the quality of clinical

trials. This topic is important in light of the existing diversity challenges that currently exist in the biomedical research field (*Valantine and Collins, 2015*).

In this study, we therefore aimed to quantify the effect of collaborations across gender combinations of junior and senior authors on the methodological quality of clinical trials. To this aim, we determined the percentage of adequately powered trials in 31,873 clinical trials published between 1974 and 2017 based on Cochrane meta-analyses. As statistical power reflects the chance of detecting a true effect, it is regarded as one of the key elements of responsible research (*Button et al., 2013*) and considered essential in reproducible clinical research (*Halpern et al., 2002*). We found that the probability of having adequate statistical power for one combination - male first author, female last author - was significantly higher than

that for the other three possible combinations. Moreover, this effect was present across countries and most medical fields.

## Results

### Statistical power and gender combinations in all clinical trials (N=31,873)

In our 31,873 trials, the number of published clinical trials with adequate statistical power (>80%) was generally low (12-13%; *Figure 1A*, left panel). The exception was the set of trials with a male first author combined with a female last author with 20.6% of outcomes adequately powered (CI 19.4–21.8). This percentage was significantly higher in comparison to the three other combinations (highest odds ratio 2.08, CI 1.87–2.30, P<0.0001). Cut-off values for adequate power set to either 70% or 90% yielded comparable results (P<0.0001; *Figure 1B*). The percentage of adequately powered trials in which the gender combination was unknown was 13.8% (CI 13.6–14.1; *Figure 1C*). Irrespective of the gender of the first author, clinical trials with female last authors had a higher statistical power compared to male last authors: 16.6% (CI 15.9–17.4) versus 12.9% (CI 12.6–13.3;

*Figure 2*). The average statistical power of clinical trials with missing gender was comparable to those with known gender combinations (*Figures 1C* and *2*). Slightly higher odds for adequately powered trials were also found in the author combination 'both males' and 'female – male (last)' in comparison to the reference group 'both females': odds ratios 1.28 (CI 1.17–1.41, P<0.0001) and 1.25 (CI 1.13–1.39, P<0.0001), respectively (*Table 1*). In the sensitivity analysis model estimates were slightly lower (relative estimate difference 2.3% to 4.8%; *Table 2*).

### Trends across countries

The world map in *Figure 3* shows the geographical distribution of the trials in our sample (based on affiliation of the first author). The percentage of trials originating from Anglosphere countries (United States, United Kingdom, Canada, Australia and New Zealand) was 46.9%; the percentage from European countries was 32.9%; and the percentage from non-western countries was 20.2% (with the top five being Turkey, Japan, India, China and Israel). European trials had lower odds of adequate statistical power compared to Anglosphere trials (odds ratio: 0.76, CI 0.71–0.81, P<0.0001; *Figure 4A*). This was also the case in trials from Non-western countries

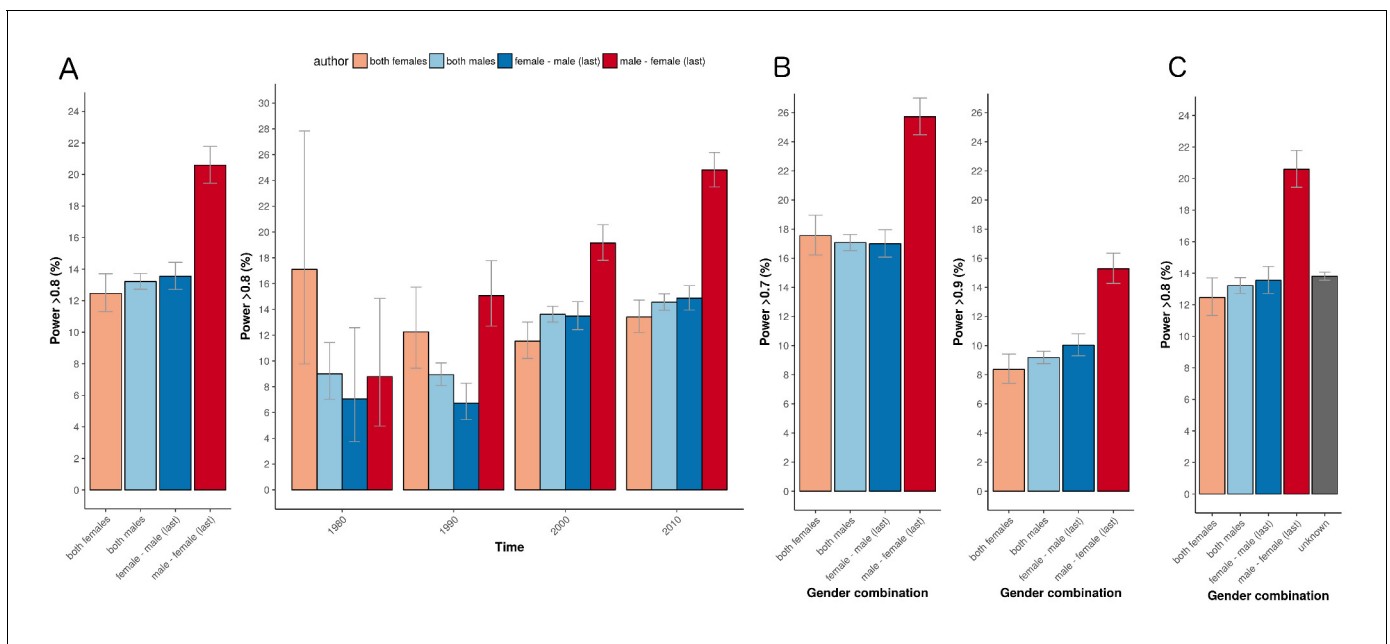

**Figure 1.** Percentage of adequately powered trials for the four different gender combinations of first and last author. (A) Percentage of trials with power > 0.8 published between 1974 and 2017 for the four gender combinations (left panel) and for four periods (1975–1985; 1985–1995; 1995–2005; >2005) during this time (right panel). (B) Percentage of trials published with power > 0.7 (*left*) and power > 0.9 (*right*) for the four gender combinations. (C) Percentage of trials with power > 0.8 for the four gender combinations, including the trials were gender could not be determined for the first and/or last author ('unknown'). Error bars represent the 95% confidence interval for proportions for all panels.
DOI: https://doi.org/10.7554/eLife.34412.002

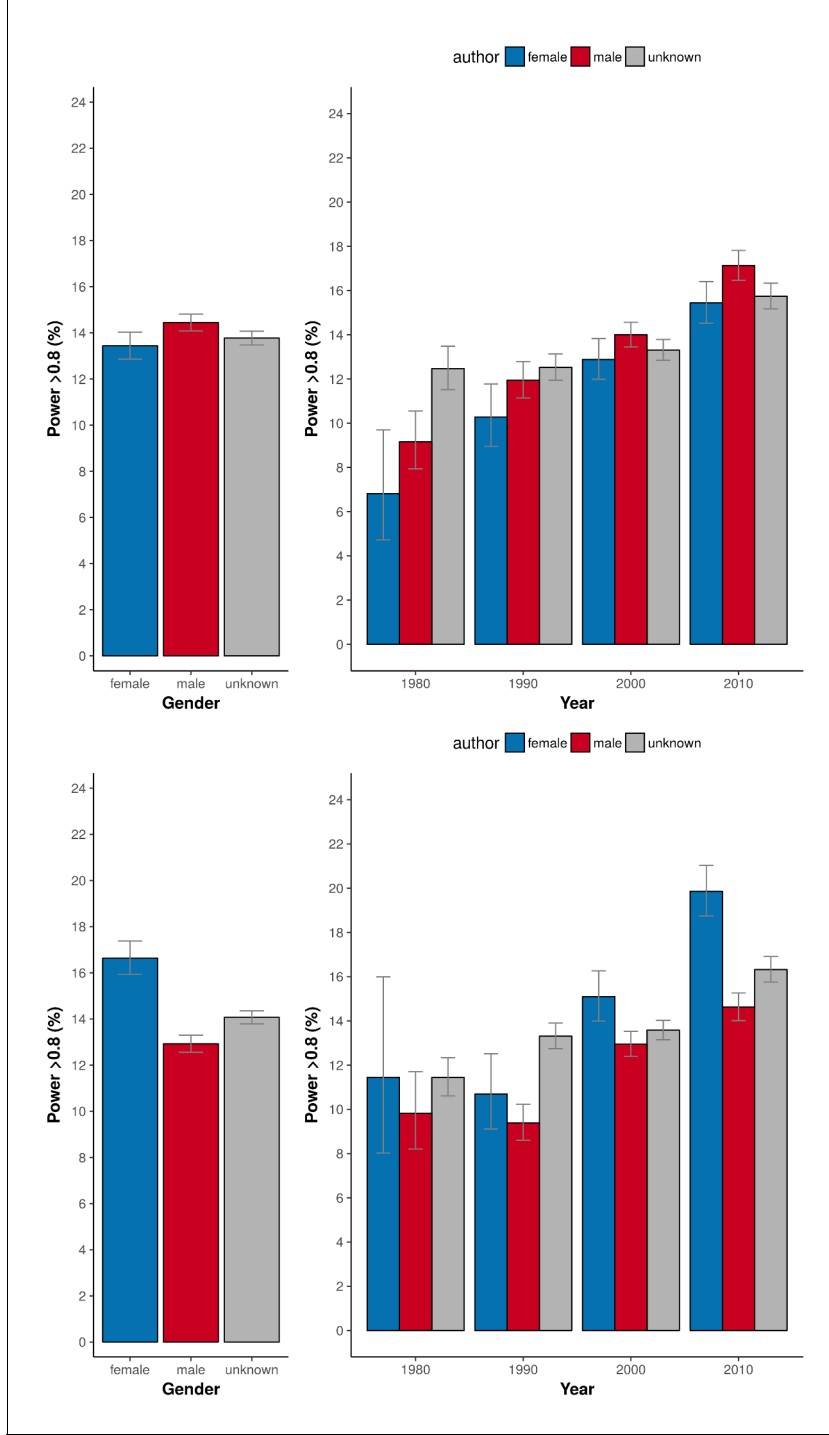

**Figure 2.** Percentage of adequately powered trials when the gender of the first and last author is male, female or unknown. *Left*: Percentage of trials with power > 0.8 plotted for the gender of the first author (top) and the last author (bottom). *Right*: Percentage of trials with power > 0.8 plotted for four periods (1975–1985; 1985–1995; 1995–2005; >2005) for the gender of the first author (top) and the last author (bottom). Error bars represent the 95% confidence interval for proportions for all panels.

DOI: https://doi.org/10.7554/eLife.34412.003

(odds ratio: 0.87, CI 0.80–0.94, P<0.0001; *Table 1*). Individual country data, from the sensitivity analysis, is provided in *Table 2*.

### Trends over time

The percentage of adequately powered trials with a male first author and a female last author increased over time, and was higher than the percentage for other combinations in the last three decades of the study (*Figure 1A*, right panel). According to a logistic regression multivariable model (see "Data analysis and statistical model") the odds ratio of adequate statistical power increased each year (odds ratio: 1.03, CI: 1.02–1.03, P<0.0001; *Figure 4B*).

### Trends across medical fields

The higher percentage of adequately powered clinical trials with a combination of a male first author and a female last author was not restricted to specific medical disciplines, although the effect sizes differed across disciplines (*Figure 5*). The medical fields with a relative low odds for adequate statistical power in general, as determined with the multivariable model, are: 'complementary medicine', 'endocrine & metabolic', 'gastroenterology & hepatology', 'genetic disorders', 'health & safety at work' and 'heart & circulation', all with significant odds ratios below 0.3 compared to the reference field allergy and intolerance (*Table 1*). The fields with most pronounced higher statistical power for male first and female last author were 'pregnancy & childbirth', 'gynaecology', 'lungs & airways', 'gastroenterology & hepatology' and 'tobacco, drugs & alcohol'. The total number of trials for each of the four gender combinations was not equally distributed. Most trials were published by the male–male author combination (*Figure 6*), and this inequality in the gender of authors was found across major medical disciplines (*Figure 6*). Nevertheless, the number of clinical trials with a male first and last author decreased from 64.8% in the period 1975–1985 (CI 61.9–67.6) to 49.0% after 2005 (CI 48.4–49.6; *Figure 7*).

### Correction for potential confounders

To correct for the potential confounders at the country level, the year of publication, and the medical discipline, logistic regression was performed. The linear combination of the variables 'author combination', 'year of publication', 'country' and 'medical discipline' explained the presence or absence of adequate statistical

**Table 1.** Model estimates for the variables fitted against adequately powered trials.

| Variables | Odds ratio | 95% CI | | Z value | P value |
|---|---|---|---|---|---|
| *Author combination* | | | | | |
| Both females | 1.00 (ref.) | | | | |
| Both males | 1.28 | 1.17 | 1.41 | 5.22 | <0.0001 |
| Female - male (last) | 1.25 | 1.13 | 1.39 | 4.29 | <0.0001 |
| Male - female (last) | 2.08 | 1.87 | 2.30 | 13.94 | <0.0001 |
| *Time* | | | | | |
| Publication year | 1.03 | 1.02 | 1.03 | 12.05 | <0.0001 |
| *Country group* | | | | | |
| Anglosphere | 1.00 (ref.) | | | | |
| Europe | 0.76 | 0.71 | 0.81 | 8.87 | <0.0001 |
| Non-western | 0.87 | 0.80 | 0.94 | 3.69 | <0.0001 |
| *Medical discipline* | | | | | |
| Allergy & intolerance | 1.00 (ref) | | | | |
| Blood disorders | 0.45 | 0.34 | 0.62 | 5.11 | <0.0001 |
| Child health | 0.47 | 0.36 | 0.61 | 5.68 | <0.0001 |
| Complementary medicine | 0.23 | 0.17 | 0.31 | 9.14 | <0.0001 |
| Consumer strategies | 0.66 | 0.41 | 1.03 | 1.80 | 0.072 |
| Dentistry & oral health | 1.05 | 0.68 | 1.59 | 0.21 | 0.832 |
| Developmental problems | 0.69 | 0.47 | 1.00 | 1.98 | 0.048 |
| Ear, nose & throat | 0.37 | 0.24 | 0.55 | 4.77 | <0.0001 |
| Effective health systems | 0.75 | 0.53 | 1.07 | 1.57 | 0.115 |
| Endocrine & metabolic | 0.29 | 0.20 | 0.42 | 6.51 | <0.0001 |
| Eyes & vision | 0.56 | 0.38 | 0.81 | 3.02 | 0.003 |
| Gastroenterology & hepatology | 0.49 | 0.38 | 0.65 | 5.07 | <0.0001 |
| Genetic disorders | 0.19 | 0.12 | 0.30 | 7.09 | <0.0001 |
| Gynaecology | 0.69 | 0.52 | 0.92 | 2.58 | 0.01 |
| Health & safety at work | 0.24 | 0.13 | 0.42 | 4.74 | <0.0001 |
| Heart & circulation | 0.29 | 0.22 | 0.39 | 8.24 | <0.0001 |
| Infectious disease | 0.61 | 0.47 | 0.80 | 3.62 | <0.0001 |
| Kidney disease | 0.80 | 0.58 | 1.12 | 1.28 | 0.201 |
| Lungs & airways | 0.35 | 0.27 | 0.46 | 7.56 | <0.0001 |
| Mental health | 0.53 | 0.40 | 0.71 | 4.40 | <0.0001 |
| Neonatal care | 0.47 | 0.34 | 0.64 | 4.68 | <0.0001 |
| Neurology | 0.56 | 0.42 | 0.74 | 4.08 | <0.0001 |
| Orthopaedics & trauma | 0.79 | 0.60 | 1.05 | 1.63 | 0.103 |
| Pain & anaesthesia | 0.64 | 0.49 | 0.84 | 3.23 | 0.001 |
| Pregnancy & childbirth | 0.58 | 0.44 | 0.77 | 3.76 | <0.0001 |
| Public health | 1.23 | 0.78 | 1.92 | 0.89 | 0.372 |
| Rheumatology | 0.75 | 0.57 | 1.00 | 2.02 | 0.043 |
| Skin disorders | 0.89 | 0.65 | 1.23 | 0.69 | 0.488 |
| Tobacco, drugs & alcohol | 0.34 | 0.26 | 0.46 | 7.39 | <0.0001 |
| Urology | 1.04 | 0.74 | 1.45 | 0.21 | 0.834 |
| Wounds | 0.36 | 0.21 | 0.61 | 3.73 | <0.0001 |

DOI: https://doi.org/10.7554/eLife.34412.004

power well in a multivariable logistic regression model ($\chi^2$*Zeng et al., 2016* = 1146.5 (degree-of-freedom 36), P<0.0001). The four author combinations were overall different from each other ($\chi^2$*Zeng et al., 2016* = 440.5 (4), P<0.0001). The model estimates are provided in *Table 1*. A sensitivity analysis with 'country' defined as individual countries rather than groups of countries did not significantly change the other variable model estimates (*Table 2*). The sensitivity model explained the presence or absence of adequate statistical power very well in a multivariable logistic regression model ($\chi^2$*Zeng et al., 2016* = 3638.6 (degree-of-freedom 101), P<0.0001). The four author combinations in the sensitivity analysis were also overall different from each other ($\chi^2$*Zeng et al., 2016* = 488.2 (4), P<0.0001).

## Discussion

The analysis of 31,873 clinical trials published between 1974 and 2017 demonstrates that adequately powered clinical trials are relatively more often published by a combination of a male first author and a female last author compared to other gender combinations. This effect was robust as the effect was present across countries and most medical fields. Even though the average statistical power was generally low, the overall percentage of adequately powered trials slightly increased over the past four decades.

In line with recent literature, (*West et al., 2013*) the absolute number of clinical trials published by female authors remained relatively low, even though it increased over time. The effects of equal representation of male and female scientists are not only important to better understand the success of collaborative efforts, but are also pressing in light of the persistent gender gap in medicine. Despite improvements, female scientists continue to face unequal pay (*Rimmer, 2017*) and funding disparities (*Shen, 2013*), and to remain underrepresented in clinical medicine in terms of the clinical faculty positions and first author publications (*Jagsi et al., 2006*; *Reed et al., 2011*), even though gains in participation have been made over the last years (*Filardo et al., 2016*). Independent of gender, the overall percentage of adequately powered clinical trials was disappointingly low, notwithstanding the fact that the practice of conducting clinical trials with low statistical power has been denounced for a long time (*Halpern et al., 2002*; *Ioannidis, 2005*). On a more positive note, the percentage of adequately powered trials did increase slightly over

the past four decades. A possible reason for this increase may be the obligation to register clinical trials (i.e., on platforms like clinicaltrials.gov). This may have caused an increase in pre-registrations and research protocols with a higher quality and commitment to the original research plan and proposed sample size.

Our results support previous reports that gender differences exist and may influence the quality of clinical trials (*Campbell et al., 2013*; *Nielsen et al., 2017*). It may also be influenced by collaboration style patterns as differences exist between men and women in mixed-sex interactions (*Balliet et al., 2011*). Firm evidence on the influence of collaborative styles is still lacking. (*Zeng et al., 2016*; *Araújo et al., 2017*) However, the impact of social behavior between clinical researchers on trial outcomes – particularly related to gender - is yet a rather unexplored area. It is important to note that not all studies have found convincing evidence for gender differences in science, (*Hyde, 2005*) for example with regard to bias (*Fanelli et al., 2017*). From our results, it could be hypothesized that collaborations between male and female researchers are beneficial with respect to cross-fertilization, team productivity and research efficacy. However, our understanding of social and gender-related factors that underlie clinical trial quality is probably still limited, which is underlined by our finding that the statistical power of trials is relatively low when both first and last author are female.

Because our analyses are based on a comprehensive body of clinical trials published over a 40-year period, across medical fields, the results provide a representative picture of the relation between gender collaborations and statistical power. Nonetheless, there are several limitations. First, we only investigated one aspect of methodological rigor. Even though statistical power is an important sign of sound trial conduct, there are other domains, including pre-post registration mismatch and other sources of bias that determine methodological rigor. These parameters, however, are more difficult to quantify. Second, gender from the first and last author could not be determined for most included clinical trials (almost 70%, see flow diagram) as not all first name records were available. However, the statistical power of trials with missing gender data was not different from the clinical trials with known gender. Third, first and last authorships only provide a relative rough proxy for junior and senior positions. The actual hierarchical relations in a clinical trial may differ

**Table 2.** Model estimates from the sensitivity analysis (with individual countries) for the variables fitted against adequately powered trials.

| Variables | Odds ratio | 95% CI | | Z value | P value |
|---|---|---|---|---|---|
| *Author combination* | | | | | |
| Both females | 1.00 (ref.) | | | | |
| Both males | 1.25 | 1.13 | 1.37 | 4.58 | <0.001 |
| Female - male (last) | 1.19 | 1.07 | 1.32 | 3.28 | 0.001 |
| Male - female (last) | 1.98 | 1.78 | 2.19 | 12.95 | <0.001 |
| *Time* | | | | | |
| Publication year | 1.02 | 1.02 | 1.03 | 14.5 | <0.001 |
| *Country* | | | | | |
| Argentina | 1.00 (ref.) | | | | |
| Australia | 0.79 | 0.52 | 1.19 | 1.12 | 0.261 |
| Austria | 1.31 | 0.84 | 2.02 | 1.19 | 0.232 |
| Bangladesh | 3.29 | 2.00 | 5.41 | 4.69 | <0.001 |
| Belgium | 0.94 | 0.61 | 1.45 | 0.29 | 0.775 |
| Brazil | 0.98 | 0.63 | 1.51 | 0.10 | 0.92 |
| Canada | 1.16 | 0.78 | 1.72 | 0.72 | 0.474 |
| Chile | 0.74 | 0.39 | 1.42 | 0.89 | 0.371 |
| China | 1.20 | 0.8 | 1.81 | 0.87 | 0.383 |
| Colombia | 1.95 | 1.17 | 3.26 | 2.55 | 0.011 |
| Costa Rica | 0.00 | 0.00 | Inf | 0.14 | 0.891 |
| Croatia | 0.47 | 0.22 | 1.03 | 1.88 | 0.06 |
| Czech Republic | 0.71 | 0.45 | 1.13 | 1.45 | 0.147 |
| Denmark | 1.24 | 0.82 | 1.87 | 1.03 | 0.303 |
| Egypt | 1.78 | 1.13 | 2.79 | 2.50 | 0.013 |
| Finland | 0.88 | 0.58 | 1.32 | 0.63 | 0.527 |
| France | 0.91 | 0.61 | 1.37 | 0.44 | 0.663 |
| Gambia | 1.05 | 0.56 | 1.99 | 0.16 | 0.87 |
| Germany | 0.90 | 0.6 | 1.34 | 0.53 | 0.593 |
| Ghana | 0.84 | 0.48 | 1.48 | 0.61 | 0.544 |
| Greece | 0.46 | 0.29 | 0.75 | 3.12 | 0.002 |
| Hong Kong | 1.37 | 0.89 | 2.11 | 1.44 | 0.15 |
| Hungary | 2.87 | 1.75 | 4.7 | 4.18 | <0.001 |
| India | 0.89 | 0.58 | 1.35 | 0.56 | 0.573 |
| Indonesia | 0.71 | 0.34 | 1.48 | 0.93 | 0.354 |
| Iran | 1.14 | 0.73 | 1.79 | 0.59 | 0.557 |
| Ireland | 0.80 | 0.49 | 1.32 | 0.87 | 0.387 |
| Israel | 0.80 | 0.51 | 1.26 | 0.98 | 0.328 |
| Italy | 1.03 | 0.69 | 1.53 | 0.15 | 0.881 |
| Japan | 0.35 | 0.22 | 0.53 | 4.83 | <0.001 |
| Jordan | 3.91 | 2.09 | 7.32 | 4.27 | <0.001 |
| Kenya | 0.42 | 0.18 | 1.00 | 1.97 | 0.049 |
| Korea | 1.56 | 1.02 | 2.39 | 2.07 | 0.038 |
| Lebanon | 1.36 | 0.75 | 2.46 | 1.01 | 0.311 |
| Malawi | 0.12 | 0.03 | 0.52 | 2.83 | 0.005 |
| Malaysia | 0.78 | 0.34 | 1.79 | 0.59 | 0.552 |

*Table 2 continued on next page*

*Table 2 continued*

| Variables | Odds ratio | 95% CI | | Z value | P value |
|---|---|---|---|---|---|
| **Author combination** | | | | | |
| Mali | 0.75 | 0.29 | 1.91 | 0.61 | 0.543 |
| Mexico | 1.07 | 0.62 | 1.85 | 0.25 | 0.8 |
| Netherlands | 0.71 | 0.47 | 1.07 | 1.62 | 0.106 |
| New Zealand | 1.28 | 0.76 | 2.14 | 0.94 | 0.349 |
| Nigeria | 1.32 | 0.70 | 2.48 | 0.87 | 0.386 |
| Norway | 0.89 | 0.56 | 1.41 | 0.49 | 0.624 |
| Pakistan | 0.93 | 0.48 | 1.83 | 0.20 | 0.844 |
| Papua New Guinea | 0.00 | 0.00 | Inf | 0.10 | 0.918 |
| Peru | 0.99 | 0.57 | 1.7 | 0.04 | 0.967 |
| Poland | 0.39 | 0.22 | 0.68 | 3.29 | 0.001 |
| Portugal | 3.17 | 1.84 | 5.45 | 4.17 | <0.001 |
| Qatar | 0.00 | 0.00 | Inf | 0.11 | 0.916 |
| Saudi Arabia | 0.54 | 0.30 | 0.98 | 2.02 | 0.043 |
| Singapore | 1.14 | 0.68 | 1.93 | 0.50 | 0.614 |
| Slovenia | 0.00 | 0.00 | Inf | 0.11 | 0.91 |
| South Africa | 1.24 | 0.79 | 1.96 | 0.93 | 0.355 |
| Spain | 1.08 | 0.71 | 1.62 | 0.35 | 0.73 |
| Sweden | 1.24 | 0.83 | 1.85 | 1.03 | 0.301 |
| Switzerland | 0.66 | 0.43 | 1.02 | 1.89 | 0.059 |
| Taiwan | 0.45 | 0.29 | 0.71 | 3.43 | 0.001 |
| Thailand | 1.53 | 0.99 | 2.37 | 1.93 | 0.053 |
| Turkey | 0.64 | 0.42 | 0.98 | 2.06 | 0.039 |
| Uganda | 1.27 | 0.56 | 2.88 | 0.58 | 0.56 |
| UK | 1.25 | 0.84 | 1.85 | 1.10 | 0.273 |
| USA | 1.42 | 0.96 | 2.10 | 1.78 | 0.076 |
| Venezuela | 5.25 | 3.22 | 8.54 | 6.67 | <0.001 |
| Vietnam | 0.00 | 0.00 | Inf | 0.12 | 0.907 |
| Zimbabwe | 1.93 | 0.90 | 4.12 | 1.70 | 0.089 |
| Other countries | 0.75 | 0.48 | 1.17 | 1.28 | 0.201 |
| **Medical discipline** | | | | | |
| Allergy & intolerance | 1.00 (ref.) | | | | |
| Blood disorders | 0.49 | 0.39 | 0.63 | 5.79 | <0.001 |
| Child health | 0.55 | 0.45 | 0.67 | 5.87 | <0.001 |
| Complementary medicine | 0.26 | 0.20 | 0.33 | 11.21 | <0.001 |
| Consumer strategies | 0.94 | 0.66 | 1.34 | 0.35 | 0.73 |
| Dentistry & oral health | 1.43 | 1.07 | 1.92 | 2.41 | 0.016 |
| Developmental problems | 0.78 | 0.58 | 1.05 | 1.64 | 0.101 |
| Ear, nose & throat | 0.51 | 0.39 | 0.68 | 4.66 | <0.001 |
| Effective health systems | 0.85 | 0.63 | 1.14 | 1.11 | 0.269 |
| Endocrine & metabolic | 0.4 | 0.30 | 0.53 | 6.58 | <0.001 |
| Eyes & vision | 0.51 | 0.38 | 0.70 | 4.27 | <0.001 |
| Gastroenterology & hepatology | 0.56 | 0.46 | 0.69 | 5.41 | <0.001 |
| Genetic disorders | 0.29 | 0.20 | 0.42 | 6.51 | <0.001 |

*Table 2 continued on next page*

*Table 2 continued*

| Variables | Odds ratio | 95% CI | | Z value | P value |
|---|---|---|---|---|---|
| *Author combination* | | | | | |
| Gynaecology | 0.82 | 0.66 | 1.01 | 1.84 | 0.066 |
| Health & safety at work | 0.54 | 0.37 | 0.79 | 3.16 | 0.002 |
| Heart & circulation | 0.34 | 0.27 | 0.43 | 9.43 | <0.001 |
| Infectious disease | 0.8 | 0.65 | 0.99 | 2.09 | 0.036 |
| Kidney disease | 0.71 | 0.55 | 0.92 | 2.59 | 0.01 |
| Lungs & airways | 0.47 | 0.38 | 0.58 | 7.04 | <0.001 |
| Mental health | 0.6 | 0.48 | 0.75 | 4.58 | <0.001 |
| Neonatal care | 0.38 | 0.29 | 0.48 | 7.81 | <0.001 |
| Neurology | 0.7 | 0.57 | 0.87 | 3.26 | 0.001 |
| Orthopaedics & trauma | 1.18 | 0.96 | 1.46 | 1.56 | 0.12 |
| Pain & anaesthesia | 0.73 | 0.60 | 0.90 | 2.92 | 0.003 |
| Pregnancy & childbirth | 0.69 | 0.55 | 0.85 | 3.40 | 0.001 |
| Public health | 1.72 | 1.24 | 2.37 | 3.27 | 0.001 |
| Rheumatology | 0.97 | 0.78 | 1.20 | 0.31 | 0.757 |
| Skin disorders | 1.26 | 0.99 | 1.59 | 1.89 | 0.058 |
| Tobacco. drugs & alcohol | 0.4 | 0.32 | 0.50 | 7.97 | <0.001 |
| Urology | 1.27 | 1.00 | 1.63 | 1.92 | 0.054 |
| Wounds | 0.8 | 0.59 | 1.08 | 1.44 | 0.15 |

DOI: https://doi.org/10.7554/eLife.34412.005

in a subset, for instance in some disciplinary fields authors are alphabetically positioned, or the persons in charge of the actual conduct of the clinical trial in daily practice are not last author on the resulting publication. Fourth, we only have included clinical trials and although these results can be extrapolated to other types of research, other research types and other academic disciplinary fields may have other unwritten rules how to determine the authors' positions on a paper. Fifth, we do not have the data of the gender of the authors between the first and last author which may influence collaboration patterns within and between research groups.

Even though adequate power in clinical trials is of vital importance, (*Ioannidis, 2014*) future studies on gender collaborations should also

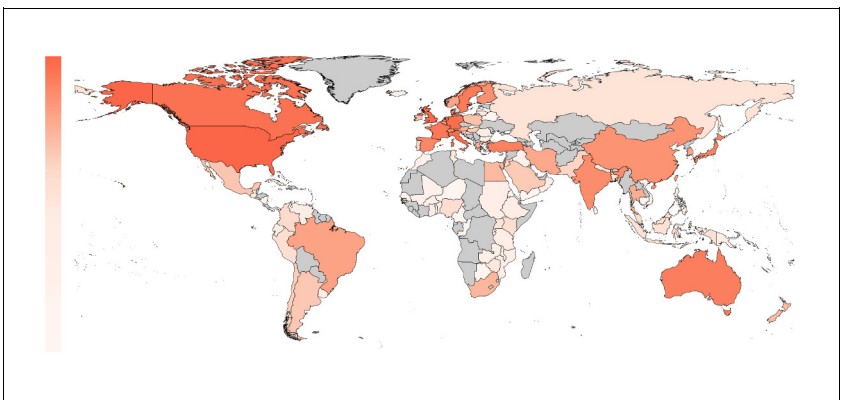

**Figure 3.** The proportion of included trials mapped per country on a white to red color scale (range: 0 – 24%). The highest proportion of first authors were affiliated with an institution in the United States. Countries not present in any affiliation are plotted in gray.

DOI: https://doi.org/10.7554/eLife.34412.006

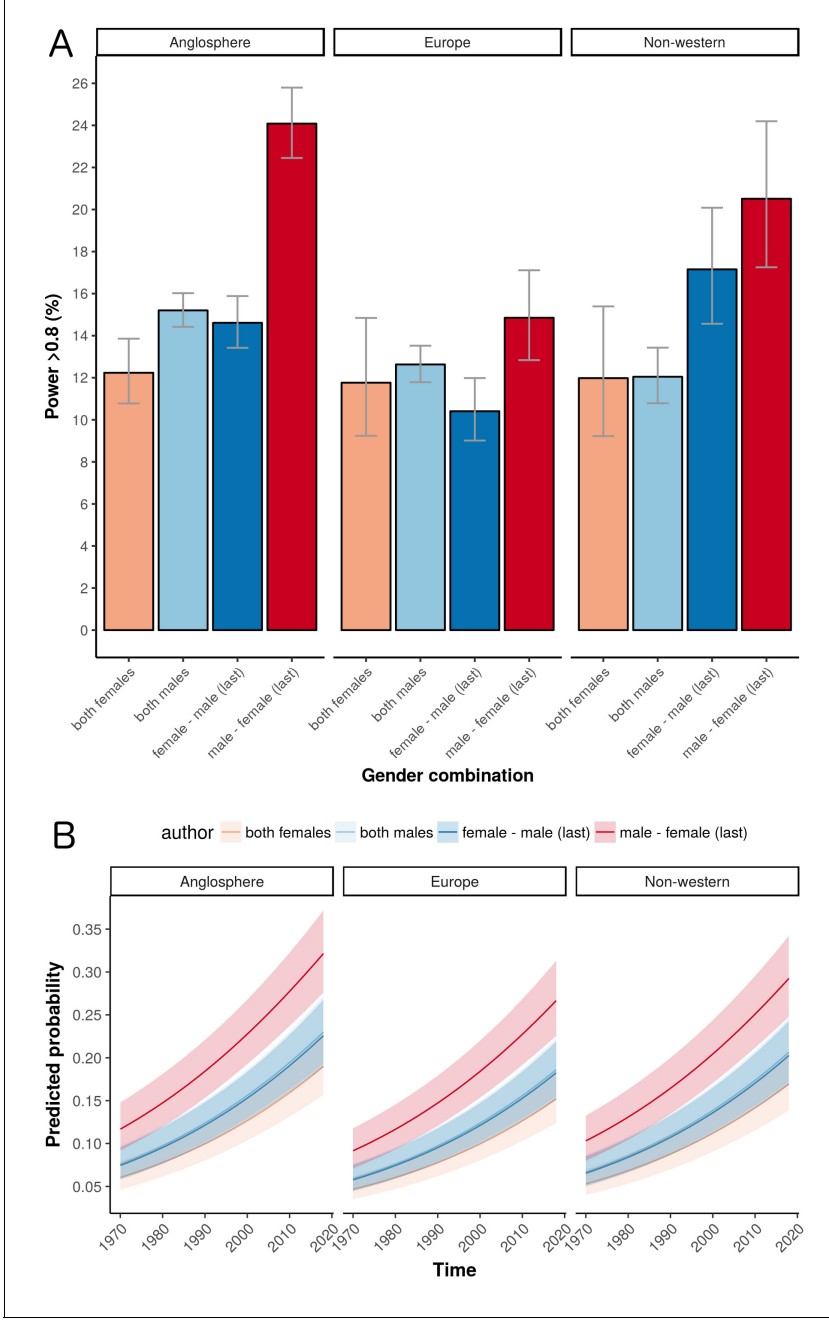

**Figure 4.** The influence of geography on the percentage of trials that are adequately powered. (A) Percentage of trials with power > 0.8 for the four gender combinations of first and last author within the three country groups. Error bars represent the 95% confidence interval for proportions. (B) A logistic regression multivariable model (see "Data analysis and statistical model" below) can be used to predict the probability that a trial will have a power above a certain value. Here the predicted probabilities that trials will have power > 0.8 are plotted as a function of year for the four gender combinations in the three country groups. The predicted probabilities are averaged across medical disciplines and plotted as mean and 95% confidence intervals.

DOI: https://doi.org/10.7554/eLife.34412.007

take other methodological outcomes into account, such as the risk of bias and deviations from the pre-registered protocol. Also, to further determine how the gender of a researcher impacts on the scientific methodological quality, a more qualitative research design would be necessary to explore on a deeper level why methodological quality of clinical trials depends on the gender of researchers and clinicians. This would include interviews and observation studies of clinical trial teams with male and female leadership positions. Our findings demonstrate the importance of gender diversity in research collaborations and emphasize the need for more prominent positions for women at senior positions in medicine (*Nature, 2013*).

## Materials and methods

### Selection of clinical trials

The selection of trials for this analysis is shown in a flow chart (*Figure 8*). Clinical trials were extracted from the Cochrane Database of Systematic Reviews. Only the subset of trials was included in the analysis where the first name of the first and last author were reported. These reviews cover all medical fields and have high quality standards and methodological rigor with elaborate search protocols, and rigorously identify and summarize comparable trials (*Jørgensen et al., 2006*). Moreover, these reviews perform meta-analyses on individual clinical trials to generate an estimated effect size of interventions. All clinical trials with a PubMed ID included in a systematic review published in the second Issue of the 2017 Cochrane Database of Systematic Reviews (CDSR) were extracted using an in-house developed, open-source Cochrane Library website parser. For each individual clinical trial, we extracted publication year, outcome estimates (odds or risk ratio, risk difference or standardized mean difference), and Cochrane's medical discipline classifications.

### Statistical power of individual clinical trials

Statistical power was assessed in clinical trials, published after 1974, which were included in a Cochrane meta-analysis with a significant overall estimate (i.e., a meta-analytic P-value of <0.05). All data and scripts are available via the Open Science Framework (WM Otte, Temporal RCT power, Open Science Framework, https://osf.io/ud2jw/. Update 17-03-04 11:19 AM). We included only significant meta-analyses to exclude bias from interventions with no proven

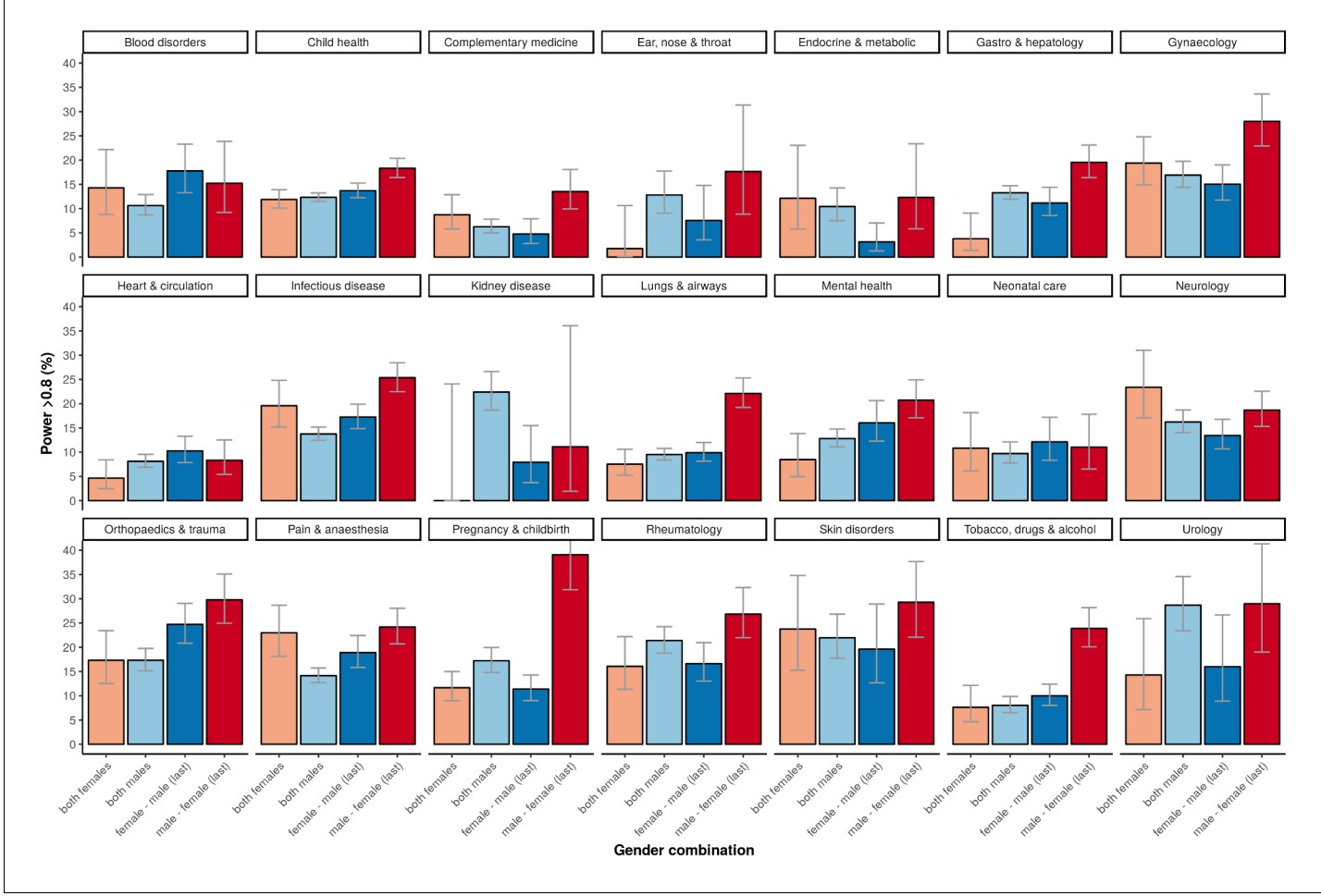

**Figure 5.** Percentage of adequately powered trials, for the four gender combinations of the first and the last author, within 21 major medical disciplines. Error bars represent the 95% confidence interval for proportions.

DOI: https://doi.org/10.7554/eLife.34412.008

effects. In other words, if a confidence interval of a meta-analysis contains 0, the point estimate of the overall effect size is not reliable nor known and may not be used to estimate the individual power of studies included in that meta-analysis. Nevertheless, inclusion of non-significant meta-analyses did not impact on our findings (data not shown).

The power for an individual clinical trial was calculated based on the sample sizes in both trial arms, using a 5% α threshold using the meta-analytic estimate as approximation of the true effect size. Trials with a statistical power lower than 80% were considered to be underpowered based on historical arguments (*Moher et al., 1994*). This cut-off is standard but also relatively arbitrary. We therefore also performed analyses using a less and more conservative cut-off of 70% and 90%, respectively. The statistical power is presented in all plots with

95% confidence intervals determined with the Wilson's score method (*Wilson, 1927*).

### Gender extraction

All included trials had multiple authors. We considered the first author of clinical trial publication as a junior researcher and the last author as a senior. This assumption will most likely reflect the hierarchal relationship in the majority of the cases. The senior author having the last position in publications has long been practiced in medicine. Typically, the person conducting the practical research, analyzing the data, and drafting the first manuscript is often the first author, while the last author is the senior research responsible for the overall oversight.

For the gender of authors, first names were extracted for the first and the last author for all included clinical trials using the online interface PubReMiner (http://hgserver2.amc.nl/cgi-bin/

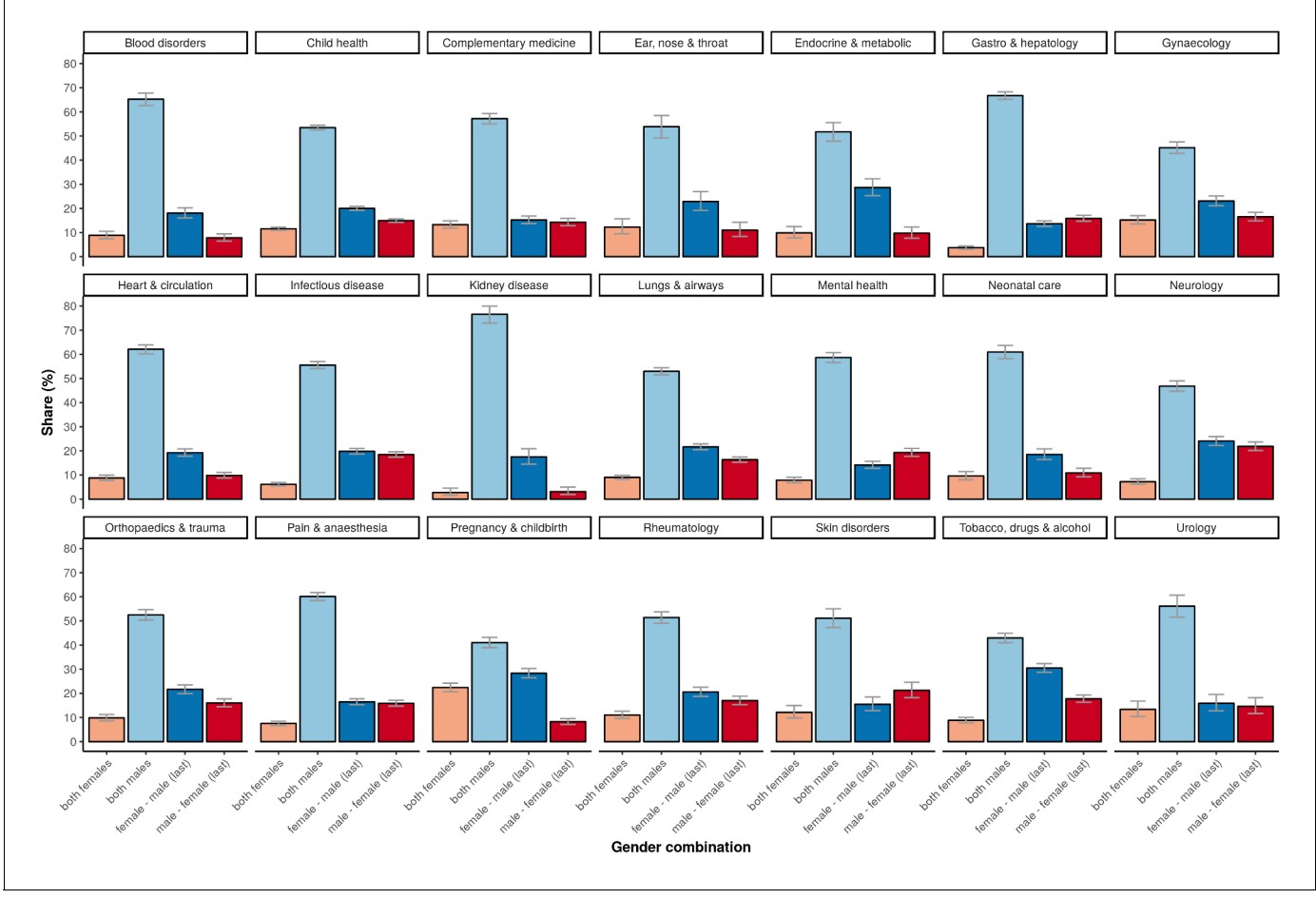

**Figure 6.** The percentage of the total number of trials underlying the four gender combinations within 21 major medical disciplines. Error bars represent the 95% confidence interval for proportions.
DOI: https://doi.org/10.7554/eLife.34412.009

miner/miner2.cgi) (*Slater, 2014*). First names were then converted to male and female probabilities with the application programming interface (API) of Genderize (http://genderize.io/). This API compares first names against a database containing over 216,000 distinct names from 79 countries and 89 languages based on millions of public profiles and their gender data in major social networks. Accuracy of female and male classification with this API, compared with open-source gender prediction tools, is excellent (*Wais, 2016*). A recent validation study reported female and male classification precisions of 95% and 98%, respectively (*Karimi et al., 2016*). Gender probabilities were dichotomized to obtain binary male/female labels. Trials with unknown gender data for either the first or last author were not included in the analysis. Missing first names caused most of the unknown genders. For some first names

no gender data was available in the gender database (<5%).

### Data analysis and statistical model
Clinical trials with adequate statistical power, more than 80%, were identified for all four combinations of the gender of first and last author (i.e. female–female, male–male, female–male and male–female).

To correct for potential cultural differences we determined the author's institutional country based on the given affiliation. We only determined this for the first author as affiliations for co-authors are added to the PubMed database only since 2014. We categorized the countries into three main groups based on prevalence. The Anglosphere countries are those where English is the main native language, the European countries, except for the United Kingdom but including Ireland, were categorized in

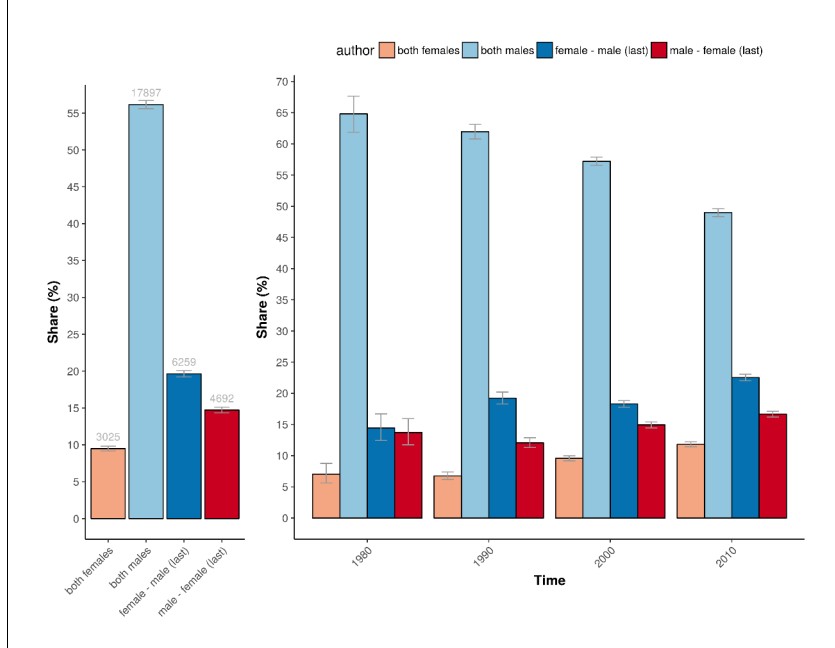

**Figure 7.** The percentage of trials for the different gender combinations and periods studied. *Left:* The number and percentage of trials underlying the power calculations for the four gender combinations. *Right*: The corresponding percentage of the total number of trials underlying the four gender combinations for the four periods studied (1975–1985; 1985–1995; 1995–2005; >2005). Error bars represent the 95% confidence interval for proportions for both panels.

DOI: https://doi.org/10.7554/eLife.34412.010

Cochrane Database of Systematic Reviews
↓
**3,111** systematic reviews with meta-analysis data
↓
**9,432** meta-analyses
↓
**6,396** significant meta-analyses
↓
**100,566** included trials, published >1974
↓
**48,423** trials with gender of first author
↓
**42,511** trials with gender of last author
↓
**31,873** trials with gender of first & last author
↓
analysis

**Figure 8.** Flow diagram of the 31,873 trials selected for analysis. Trials were analyzed if published after 1974, included in a significant meta-analysis in a systematic review and gender data was extractable for both the first and the last author.

DOI: https://doi.org/10.7554/eLife.34412.011

another group. The remaining countries were labeled as Non-western.

We classified the trials using the 21 standard Cochrane major medical discipline classifications. To exclude selection bias, statistical power was determined for all clinical trials with missing gender data. To ascertain that results were not due to disproportionate female underrepresentation in older trials, the absolute number of clinical trials for the four different gender combinations was also calculated.

The data were modeled with logistic regression. In this model the log odds of the dichotomous outcome variable, namely trial 'adequate power', was modeled as a linear combination of predictor variables. We used the *glm* function in R software version 3.2.0. The variable 'author combination' was added as a factor to the model, with the author combination 'both females' as reference group. The three covariates included were 'publication year', 'country' and 'medical field'. The model fit was investigated with the significance of the overall model. This $\chi$ (*Zeng et al., 2016*) test determines whether the model with predictors fits significantly better than a so called null model with just an intercept. The 95% confidence intervals for the estimated coefficients were determined with the profiled log-likelihood function (*Venzon and Moolgavkar, 1988*).. The estimates were exponentiated to interpret them as odds-ratios. The overall effect of 'author combination' in the model was tested with the Wald test. We determined the model's predicted probabilities and their 95% confidence intervals over time. We considered a P-value<0.005 as significant (*Benjamin et al., 2018*). We performed a sensitivity analysis with the 'country' variable not specified into three main categories but into individual country categories, if a minimal of fifty entries per country were present.

### Data sharing

Open-source code to reproduce our processing pipeline is provided via the Open Science Framework (WM Otte, Temporal RCT power, Open Science Framework, https://osf.io/ud2jw/. Update 17-03-04 11:19 AM). Data extraction from the Cochrane Database of Systematic Reviews requires full text access.

### Acknowledgements

We appreciate the valuable input and suggestions by the reviewers. We gratefully acknowledge the possibility that the all-male author list

may have negatively affected the scientific quality of our work.

**Willem M Otte** is in the Biomedical MR Imaging and Spectroscopy Group, Center for Image Sciences and the Department of Child Neurology, Brain Center Rudolf Magnus, University Medical Center Utrecht/Utrecht University, Utrecht, The Netherlands

https://orcid.org/0000-0003-1511-6834

**Joeri K Tijdink** is in the Department of Philosophy, VU University, Amsterdam, The Netherlands

https://orcid.org/0000-0002-1826-2274

**Paul L Weerheim** is in the Biomedical MR Imaging and Spectroscopy Group, Center for Image Sciences, University Medical Center Utrecht/Utrecht University, Utrecht, Netherlands

**Herm J Lamberink** is in the Biomedical MR Imaging and Spectroscopy Group, Center for Image Sciences and the Department of Psychiatry, Brain Center Rudolf Magnus, University Medical Center Utrecht/Utrecht University, Utrecht, The Netherlands

https://orcid.org/0000-0003-1379-3487

**Christiaan H Vinkers** is in the Department of Psychiatry, Brain Center Rudolf Magnus, University Medical Center Utrecht/Utrecht University, Utrecht, The Netherlands

c.h.vinkers@umcutrecht.nl

https://orcid.org/0000-0003-3698-0744

*Author contributions:* Willem M Otte, Conceptualization, Formal analysis, Supervision, Funding acquisition, Investigation, Methodology, Writing—original draft, Writing—review and editing; Joeri K Tijdink, Conceptualization, Investigation, Writing—original draft, Writing—review and editing; Paul L Weerheim, Formal analysis, Methodology; Herm J Lamberink, Conceptualization, Formal analysis, Methodology, Writing—original draft, Writing—review and editing; Christiaan H Vinkers, Conceptualization, Supervision, Funding acquisition, Investigation, Methodology, Writing—original draft, Project administration, Writing—review and editing

*Competing interests:* The authors declare that no competing interests exist.

Funding

| Funder | Grant reference number | Author |
|---|---|---|
| ZonMw | 445001002 | Willem M Otte Joeri K Tijdink Herm J Lamberink Christiaan H Vinkers |
| VENI | 016.168.038 | Willem M Otte |

The funders had no role in study design, data collection and interpretation, or the decision to submit the work for publication.

**Decision letter and Author response**
Decision letter https://doi.org/10.7554/eLife.34412.015
Author response https://doi.org/10.7554/eLife.34412.016

## Additional files

### Data availability

Data and scripts are available via the Open Science Framework (https://osf.io/ud2jw/) Parsing of genders can be done via genderize.io.

The following previously published dataset was used:

| Author(s) | Year | Dataset URL | Database, license, and accessibility information |
|---|---|---|---|
| Willem M. Otte | 2017 | https://osf.io/ud2jw/ | Available at Open Science Framework |

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
