## [Decision Letter]

Thank you for submitting your article "Opposite sexes attract: male-female collaborations and the statistical power of 31,873 clinical trials" to *eLife* for consideration as a Feature Article. Your article has been reviewed by three peer reviewers, and the evaluation has been overseen by myself as the *eLife* Features Editor. The following individuals involved in review of your submission have agreed to reveal their identity: Andreas Neef (Reviewer #1).

The reviewers have discussed the reviews with one another and I have drafted this decision to help you prepare a revised submission

Summary:

The study presented is an analysis of over 30,000 clinical trials to assess whether gender pairings of first and last authors are associated with sufficient statistical power. The study employs a large data set and uses the API Genderize to assign gender. The authors find that overall a relatively low number of studies had sufficient statistical power (12-13%). Similar to other studies looking at gender pairings, the authors find that most studies are comprised of male first-male last authors pairings. Over the forty years analyzed, the authors observed a gradual increase in the number of female first and last authors in clinical trials. When assessed for sufficient power, studies comprised of a first male author and female last author were significantly more prevalent compared to other gender pairings.

However, a number of concerns need to be addressed before the article can be accepted for publication. In particular, there is a need for more transparency with respect to data inclusion/exclusion, and the authors need to pay more careful attention to potential confounders and possible selection bias (see points 1-8 below). The Discussion section and the references cited also need extensive attention (points 9 and 10). The Title also needs to be revised (point 11)

Essential revisions:

Confounders: I see three possible confounders that may bias the results of this study. Time, discipline-specific traditions and country variation. I will briefly touch upon each of these possible confounders below and suggest a suitable solution.

1) As mentioned by the authors, the underrepresentation of female authors in older trials may bias the results given the general increase in the percentage share of sufficiently powered trials over time. The authors account for this in Figure 1 (right panel) by dividing the results into periodical bins. But what about the possible variation within these ten-year periodical bins? A more refined approach could be to use publication year as a covariate in a binary logistic regression (I return to this).

2) Some medical disciplines may be more likely to conduct sufficiently powered trials than others, irrespective of author-gender composition. It is difficult infer whether this is the case from Figure 2. But four of the disciplines with the most pronounced higher statistical power for studies with female last authors (gynaecology, pregnancy and childbirth and tobacco and drugs and infectious diseases) also have generally good female author representation as presented in Figure 6. Consequently, the general difference detected in your study may possibly be explained by a higher general proportion of women authors in disciplines that are have relatively high general proportions of sufficiently powered studies. Or alternatively, the actual differences may possibly be larger than what you report, if the areas where women authors are well-represented are less likely to reach sufficient statistical power. Again, a more refined approach would be to do a binary logistic regression where you include covariates for the average proportion of female last authors and female authors per discipline.

3) Your study does not adjust for country-variations, which makes sense given your primary emphasis on descriptive statistics. My worry is that women's general representation as trial authors may be higher in countries that score relatively high with respect to proportion of sufficiently powered studies. If, for instance, North America (Canada and US) and Northern Europe have good numbers of women trial authors, while also being "top-performers" with respect to sufficiently powered studies, this may bias your results. Here you could also calculate a covariate adjusting for women's proportional participation as authors in a given geographic region (North America, North-Western Europe, South-East-Asia, Oceania etc.) and use this in a binary log reg.

To briefly summarize I suggest that you adjust for these possible confounders in a binary logistic regression with the outcome variable: 0=insufficiently powered 1= sufficiently powered. The regression could include your author-composition categories + the following covariates: publication year, proportion of women authors in the discipline, proportion of women last authors in the discipline, proportion of women authors in the geographical region where the trial took place, and proportion of women last authors in the geographical region where the trial took place.

4) Possible selection bias: In subsection “Statistical power of individual clinical trials” it is mentioned that you only included significant meta-analyses to exclude bias from interventions with no proven effects? I miss some sort of justification for this choice? Why would interventions with no proven effect bias your results? It would also be useful to know whether the gender composition of authors contributing to studies with no effects is similar to that of the analysed sample. Is female authors' representation higher in the sample with no proven effects relative to the sample used in the analysis?

5) Transparency with respect to data inclusion and exclusion: It would be helpful to have some sort of flow-diagram specifying the data exclusion steps. (1) How many trials did you start out with? (2) were you able to extract the necessary data from all eligible clinical trials identified in Cochrane? (3) How many studies were excluded due to statistical insignificance? (4) How many studies were excluded due to missing gender specification?

6) It would aid interpretation if the breakdown of numbers (studies excluded, numbers per category of pairing, those with sufficient statistical power….) were provided in a Table or in a Figure (similar to how presented in Figure 7).

7) Please also ensure that the following information is available in the manuscript:

- information on the definition of confidence intervals (was bootstrap used?)

- information on the statistical test applied. chi^{2}-test is mentioned, to me it is unclear if multiple comparison correction was used and why the continuity correction has to be used when actually _numbers_ of studies are tested. Also unclear is whether the bimodal distribution of statistical power in studies (see Button et al., 2013) would still have us expect a normal distribution (or hypergeometric) of the number of cases in each class. Even more concerning is the potential internal structure in the dataset: how many different seniors carry the effect? etc.

So to address all this within one analysis: Why not use randomized gender label swapping?

8) A more elaborate description of the gender disambiguation would also strengthen the study. In subsection “Gender extraction” you mention that "gender probabilities were dichotomized to obtain binary male/female labels". I presume this means that your gender algorithm provides numerical name-to-gender accuracy estimations? (please describe in more detail how the algorithm works). If yes, what threshold did you use? It would also strengthen the study if you could manually check a sub-sample of authors to verify the accuracy of the algorithm with the chosen threshold.

9) A number of reference (e.g., Helmer et al., 2017; Wuchty, Jones and Uzzi, 2007; Jones, Wuchty and Uzzi, 2008; Woolley et al., 2010 and Rhoten and Pfirman, 2007) are quoted out of context. Woolley et al., 2010 does not focus on research teams: rather, it finds that the collective intelligence of teams increase with women's representation, not with gender diversity. Rhoten and Pfirman, 2007 finds that women (not gender diverse teams) are more likely to do interdisciplinary research.

10) The Discussion section needs to be completely rewritten to focus on:i) how the results in the present manuscript relate to the existing literature (including possible explanations for the results).ii) the shortcomings and caveats associated with the present approach (e.g., just one type of article, just one area of science, just the first and last authors); and only first/last author). iii) open questions for future research.

As mentioned above, please ensure that any papers cited are directly relevant to the subject being discussed.

11) The Title needs to be revised to better reflect the content of the article. Also, please note that punctuation like colons, semi-colons and hypens/dashes are not allowed in the title of *eLife* articles.

[Editors' note: further revisions were requested prior to acceptance, as described below.]

Thank you for submitting a revised version of your manuscript "Adequate statistical power of clinical trials is associated with the combination of a male first author and a female last author". The revised version has been reviewed again by one of the referees who reviewed the original version. Once you have satisfactorily addressed the comments from this reviewer (see below) and some editorial comments from myself (also below), we should be able to accept your article.

The authors have done a good job addressing my main concerns about confounders. The logistic regression analysis has strengthened the study a lot. I think the paper is ready for publication with some revisions.

1) The authors used categorical values with three values to adjust for geo-cultural variation. I am not convinced that this is sufficient, and given the size of the data-set, I suggest that they run the same log-reg model with dummy for all countries to ensure that potential confounders at the country level are sufficiently adjusted for. If this model lead to the same results, they can choose to report the model as it is given now.

2) I suggest that the authors revise the Results section so that descriptive findings are presented first, followed by the outcomes of the logistic regression analysis. Mixing the results from the different approaches is somewhat confusing. The log-reg should be used to validate the descriptive results.

3) Regarding the Discussion section: I am still not convinced by the arguments about gender differences in collaboration style patterns. It is difficult for me to understand how your research adds to discussions of collaboration style. Collaboration style is a question of process, your study can only contribute to the understanding of outcomes? I would argue that your study contributes to the emerging literature about how gender diversity may influence research outcomes. Here are some references for other relevant studies addressing this issue (one of which – Campbell et al., 2013 – is already cited):

# Valantine and Collins (2015) (This paper could work quite well as a motivation for your manuscript])

# Nielsen et al., (2017)

# Nielsen et al., (2017)

# Joshi, (2014)

# Campbell et al., (2013)

4) Likewise, I am not convinced by the speculative argument that pairs of female first and last authors have less powered studies due to women seniors being more critical of female employees. How would being critical towards female coauthors lower statistical power? I can't see the link, and I suggest that you skip this reflection entirely.

5) Woolley et al., 2010 is not about research teams, but teams in general.

6) I feel the Title would read better if the fourth word was changed from "of" to "in" so that the title read as follows:

Adequate statistical power in clinical trials is associated with the combination of a male first author and a female last author.

However, please feel free to keep the present title if you wish.

7) I feel that the abstract and introduction would read better as follows:

Abstract

“Clinical trials have a vital role in ensuring the safety and efficacy of new medical treatments and interventions. A key characteristic of a clinical trial is its statistical power. Here we investigate whether the statistical power of a trial is related to the gender of first and last authors on the paper reporting the results of the trial. Based on an analysis of 31,873 clinical trials published between 1974 and 2017, we find that adequate statistical power was most often present in clinical trials with a male first author and a female last author (20.6%, 95% confidence interval 19.4-21.8%), and that the difference between this figure and the figure for other gender combinations was significant (12.5-13.5%; P < 0.0001). The absolute number of female authors in clinical trials also increased gradually over time, with the percentage of female last authors rising from 20.7% (1975-85) to 28.5% (after 2005). Our results demonstrate the importance of gender diversity in research collaborations and emphasize the need to increase the number of women in senior positions in medicine.”

Introduction section

“Clinical trials are complex projects that often involve collaborations between researchers who have different areas of expertise and different levels of seniority. The statistical power of a clinical trial reflects the chance of detecting a true effect, so adequate statistical power is regarded as one of the key elements of responsible research [8] and is considered essential in reproducible clinical research [9]. However, there is an increasing awareness that many clinical trials have systematic methodological flaws, including a lack of adequate statistical power, and that their results may be biased, exaggerated, and difficult to reproduce [1].

Male and female researchers differ in their collaborative strategies in ways that depend on their levels of expertise and whether they have a junior or senior position [4,5]. There are also indications that mixed gender research groups may make the best use of personal knowledge and skills [6,7], but possible relationships between the gender balance of collaborations and the quality of clinical trials – as measured by their statistical power – have not been investigated. In this study, we examined 31,873 clinical trials published between 1974 and 2017 to see if there was any relation between the gender of the first and last authors and statistical power. We found that the probability of having adequate statistical power for one combination – male first author, female last author – was significantly higher than that for the other three possible combinations. Moreover, this effect was present across countries and most medical fields.”

Please feel free to revise the Abstract and Introduction section further, but please try to avoid undue overlap between the two. Also, please note that my suggestions would mean that the following sentence (and the references therein) would be deleted: "Indeed, collaborations on trial design, conduct and report are important, and the publications resulting from team efforts and multi-university research teams are more often cited and have more impact [2,3]."

8) Textual changes and clarifications:

# Please list all the Anglophone in parenthesis the first time the word Anglophone is used.

# Please clarify in the UK and Ireland are included with the Anglophone or European countries.

# Please list the five biggest non-Western countries in parenthesis the first time the phrase non-Western countries is used.

# In the third paragraph of the Discussion, please reword the passage "It is important to note that gender assumptions are not black and white" to be more precise.

# In the fourth paragraph of the Discussion section, please replace the first two sentences ("Our analyses […] statistical power") with one shorter sentence.

9) Figures

To help the presentation of the article, please do the following:

# combine Figure 1, Figure 2 and Figure 3 into one figure with three panels.

# combine Figure 6 and Figure 7 into one figure with two panels.

# please add a colour scale to the present Figure 5.

# please expand the caption for the present Figure 7 to better explain what has been measured and what is being predicted.

---

## [Author Response]

Summary:The study presented is an analysis of over 30,000 clinical trials to assess whether gender pairings of first and last authors are associated with sufficient statistical power. The study employs a large data set and uses the API Genderize to assign gender. The authors find that overall a relatively low number of studies had sufficient statistical power (12-13%). Similar to other studies looking at gender pairings, the authors find that most studies are comprised of male first-male last authors pairings. Over the forty years analyzed, the authors observed a gradual increase in the number of female first and last authors in clinical trials. When assessed for sufficient power, studies comprised of a first male author and female last author were significantly more prevalent compared to other gender pairings.However, a number of concerns need to be addressed before the article can be accepted for publication. In particular, there is a need for more transparency with respect to data inclusion/exclusion, and the authors need to pay more careful attention to potential confounders and possible selection bias (see points 1-8 below). The Discussion section and the references cited also need extensive attention (points 9 and 10). The Title also needs to be revised (point 11).

We thank the editor and all reviewers for their thoughtful and constructive feedback. Our feedback is provided below.

Essential revisions:[…] To briefly summarize I suggest that you adjust for these possible confounders in a binary logistic regression with the outcome variable: 0=insufficiently powered 1= sufficiently powered. The regression could include your author-composition categories + the following covariates: publication year, proportion of women authors in the discipline, proportion of women last authors in the discipline, proportion of women authors in the geographical region where the trial took place, and proportion of women last authors in the geographical region where the trial took place.

We agree with the reviewers that the factors ‘study year’, ‘country of author’ and ‘medical discipline’ may affect the association between ‘author type’ and the dependent variable ‘sufficiently powered’. For instance, as seen in Figure 1, there appears to be a change in adequately powered trials over time.

We re-analyzed the data with logistic regression (with outcome variable: 0=insufficiently powered 1= sufficiently powered) where ‘study year’, ‘country of author’ and ‘medical discipline’ were included as a covariates. We have added Table 1 with the corresponding odds ratio’s, 95% confidence intervals and P-values to the revised manuscript. We have also added predicted probability plots, with the 95% prediction intervals to help understand the model.

4) Possible selection bias: In subsection “Statistical power of individual clinical trials” it is mentioned that you only included significant meta-analyses to exclude bias from interventions with no proven effects? I miss some sort of justification for this choice? Why would interventions with no proven effect bias your results? It would also be useful to know whether the gender composition of authors contributing to studies with no effects is similar to that of the analysed sample. Is female authors' representation higher in the sample with no proven effects relative to the sample used in the analysis?

In the frequentist framework we cannot infer information from the point estimate of a meta-analysis if the confidence interval contains 0. As we do not know the true effect size (which could also be 0) we can also not reliable estimate the statistical power, as this requires an estimation of the true effect size. Even so, inclusion of non-significant meta-analyses did not change any of the results. We have added text and clarification in the article.

We have added data from the trials where we have no gender information. The average power for those studies is very similar to the categories ‘both females/both males/female-male (last)’, indicating that our data is not biased by these ‘unknowns’ (see Figure 7).

5) Transparency with respect to data inclusion and exclusion: It would be helpful to have some sort of flow-diagram specifying the data exclusion steps. (1) How many trials did you start out with? (2) were you able to extract the necessary data from all eligible clinical trials identified in Cochrane? (3) How many studies were excluded due to statistical insignificance? (4) How many studies were excluded due to missing gender specification?

We have constructed a flow diagram (Figure 1) and additional descriptive figures in the revised manuscript. Furthermore, we explain briefly why our gender identification software was unable to detect gender in almost 70% of the included trials (Discussion section).

6) It would aid interpretation if the breakdown of numbers (studies excluded, numbers per category of pairing, those with sufficient statistical power….) were provided in a Table or in a Figure (similar to how presented in Figure 6).

These numbers are now given in the flow diagram (Figure 8).

7) Please also ensure that the following information is available in the manuscript:- information on the definition of confidence intervals (was bootstrap used?)

All confidence intervals are 95% confidence intervals.

We have added this to the manuscript. All confidence intervals are 95% confidence intervals. The CI’s for the proportions are determined with the standard R function prop.test from the default ‘stats’ package (https://stat.ethz.ch/R-manual/R-devel/library/stats/html/prop.test.html). The confidence intervals for the odds ratios from the logistic regression model are determined with the confint function in R from the ‘stats’ package (http://stat.ethz.ch/R-manual/R-devel/library/stats/html/confint.html).

- information on the statistical test applied. chi^{2}-test is mentioned, to me it is unclear if multiple comparison correction was used and why the continuity correction has to be used when actually _numbers_ of studies are tested. Also unclear is whether the bimodal distribution of statistical power in studies (see Button et al., 2013) would still have us expect a normal distribution (or hypergeometric) of the number of cases in each class. Even more concerning is the potential internal structure in the dataset: how many different seniors carry the effect? etc.So, to address all this within one analysis: Why not use randomized gender label swapping?

As we changed the statistical testing to logistic regression the statistical analysis with chi-square is removed. We only fitted one model in our current manuscript version. We have lowered the α cutoff threshold to 0.005, based on a recent discussion on the too liberal threshold of 0.05 (Benjamin et al., 2018).

8) A more elaborate description of the gender disambiguation would also strengthen the study. In subsection “Gender extraction” you mention that "gender probabilities were dichotomized to obtain binary male/female labels". I presume this means that your gender algorithm provides numerical name-to-gender accuracy estimations? (please describe in more detail how the algorithm works). If yes, what threshold did you use? It would also strengthen the study if you could manually check a sub-sample of authors to verify the accuracy of the algorithm with the chosen threshold.

The gender extraction based on first names with genderize.io has the highest accuracy of current methods available at the moment. Comparison and validation work is reported in this paper (Wais, (2016)). The genderize.io method is compared to the method of West et al., (2013) and Larivière et al., (2013). The area-under-the ROC curve – a general measure of predictiveness – for the genderize.io software is 0.927. The predicted power is thus excellent. Another study which validated the high validity of generize.io gender prediction is the work by Fell and König, (2016). Yet another study, by Topaz et al., on gender predictions in mathematical sciences, report an accuracy of 97.5% of generize.io (Topaz and Sen, (2016)) The generize.io software provides a probability for a first name being male or female (summing to 1.0). We labeled first names based on the highest gender probability.

9) A number of references (e.g., Helmer et al., 2017; Wuchty, Jones and Uzzi, 2007; Jones, Wuchty and Uzzi, 2008; Woolley et al., 2010 and Rhoten and Pfirman, 2007) are quoted out of context. Woolley et al., 2010 does not focus on research teams: rather, it finds that the collective intelligence of teams increase with women's representation, not with gender diversity. Rhoten and Pfirman, 2007 finds that women (not gender diverse teams) are more likely to do interdisciplinary research.

We appreciate the feedback. We have adjusted the references and text to fit the context better. As for West, Jacquet and King, 2013, we think the referee was actually discussing another paper (Araujo, Araujo and Moreira, 2017). This article does show that when an article has more collaborators, the chance of having interdisciplinary collaboration is higher for female scientists. We believe that our use of this reference is justified, stating that “our results do support previous findings that gender differences exist in collaboration style patterns.”

10) The Discussion section needs to be completely rewritten to focus on:i) how the results in the present manuscript relate to the existing literature (including possible explanations for the results).ii) the shortcomings and caveats associated with the present approach (eg, just one type of article, just one area of science, just the first and last authors); and only first/last author). iii) open questions for future research.As mentioned above, please ensure that any papers cited are directly relevant to the subject being discussed.

We have rewritten the Discussion section, addressed the abovementioned shortcomings and put a focus on future research.

11) The Title needs to be revised to better reflect the content of the article. Also, please note that punctuation like colons, semi-colons and hypens/dashes are not allowed in the title of eLife articles.

We have changed the title to “Adequate statistical power of clinical trials is associated with the combination of a male first author and a female last author.”

[Editors' note: further revisions were requested prior to acceptance, as described below.]

The authors have done a good job addressing my main concerns about confounders. The logistic regression analysis has strengthened the study a lot. I think the paper is ready for publication with some revisions.1) The authors used categorical values with three values to adjust for geo-cultural variation. I am not convinced that this is sufficient, and given the size of the data-set, I suggest that they run the same log-reg model with dummy for all countries to ensure that potential confounders at the country level are sufficiently adjusted for. If this model lead to the same results, they can choose to report the model as it is given now.

Table 2 is added to the manuscript with individual countries as potential confounder. The main effect remains and is not influenced by individual countries. We have also addressed this in the Results section.

2) I suggest that the authors revise the Results section so that descriptive findings are presented first, followed by the outcomes of the logistic regression analysis. Mixing the results from the different approaches is somewhat confusing. The log-reg should be used to validate the descriptive results.

This is a valuable suggestion. We have changed the Results section; the section “model performance” was renamed “correction for potential confounders” and moved to the end of the Results section.

3) Regarding the Discussion section: I am still not convinced by the arguments about gender differences in collaboration style patterns. It is difficult for me to understand how your research adds to discussions of collaboration style. Collaboration style is a question of process, your study can only contribute to the understanding of outcomes? I would argue that your study contributes to the emerging literature about how gender diversity may influence research outcomes. Here are some references for other relevant studies addressing this issue (one of which – Campbell et al., 2013 – is already cited):# Valantine and Collins, (2015) (This paper could work quite well as a motivation for your manuscript)# Nielsen et al., (2017)# Nielsen et al., (2017)# Joshi, (2014)# Campbell et al., (2013)

We have added your suggested references to further improve the manuscript by adding text to the discussion. Now we state that the results suggest that gender differences may alter research outcomes. See the Discussion section.

4) Likewise, I am not convinced by the speculative argument that pairs of female first and last authors have less powered studies due to women seniors being more critical of female employees. How would being critical towards female coauthors lower statistical power? I can't see the link, and I suggest that you skip this reflection entirely.

We have removed the speculative arguments (Discussion section).

5) Woolley et al., 2010 is not about research teams, but teams in general.

We changed the sentence in the Introduction section to: “There are indications that mixed gender teams may make the best use of personal knowledge and skills, an effect also reported in a scientific research context.”

6) I feel the Title would read better if the fourth word was changed from "of" to "in" so that the title read as follows:Adequate statistical power in clinical trials is associated with the combination of a male first author and a female last author.However, please feel free to keep the present title if you wish.

Thank you for the suggestion, we changed the Title accordingly.

7) I feel that the abstract and introduction would read better as follows:Abstract“Clinical trials have a vital role in ensuring the safety and efficacy of new medical treatments and interventions. A key characteristic of a clinical trial is its statistical power. Here we investigate whether the statistical power of a trial is related to the gender of first and last authors on the paper reporting the results of the trial. Based on an analysis of 31,873 clinical trials published between 1974 and 2017, we find that adequate statistical power was most often present in clinical trials with a male first author and a female last author (20.6%, 95% confidence interval 19.4-21.8%), and that the difference between this figure and the figure for other gender combinations was significant (12.5-13.5%; P < 0.0001). The absolute number of female authors in clinical trials also increased gradually over time, with the percentage of female last authors rising from 20.7% (1975-85) to 28.5% (after 2005). Our results demonstrate the importance of gender diversity in research collaborations and emphasize the need to increase the number of women in senior positions in medicine.”Introduction section“Clinical trials are complex projects that often involve collaborations between researchers who have different areas of expertise and different levels of seniority. The statistical power of a clinical trial reflects the chance of detecting a true effect, so adequate statistical power is regarded as one of the key elements of responsible research [8] and is considered essential in reproducible clinical research [9]. However, there is an increasing awareness that many clinical trials have systematic methodological flaws, including a lack of adequate statistical power, and that their results may be biased, exaggerated, and difficult to reproduce [1].Male and female researchers differ in their collaborative strategies in ways that depend on their levels of expertise and whether they have a junior or senior position [4,5]. There are also indications that mixed gender research groups may make the best use of personal knowledge and skills [6,7], but possible relationships between the gender balance of collaborations and the quality of clinical trials – as measured by their statistical power – have not been investigated. In this study, we examined 31,873 clinical trials published between 1974 and 2017 to see if there was any relation between the gender of the first and last authors and statistical power. We found that the probability of having adequate statistical power for one combination – male first author, female last author – was significantly higher than that for the other three possible combinations. Moreover, this effect was present across countries and most medical fields.”Please feel free to revise the Abstract and Introduction section further, but please try to avoid undue overlap between the two. Also, please note that my suggestions would mean that the following sentence (and the references therein) would be deleted: "Indeed, collaborations on trial design, conduct and report are important, and the publications resulting from team efforts and multi-university research teams are more often cited and have more impact [2,3]."

Many thanks for improving the readability of the text. The Abstract is completely incorporated, and we have added some results in the Abstract. The Introduction section was also amended. We have added one line to the Acknowledgments section in appreciation of your and the reviewers’ suggestions.

8) Textual changes and clarifications:# Please list all the Anglophone in parenthesis the first time the word Anglophone is used.

We removed the mixed of anglophone and anglosphere from the text and kept anglosphere. These are formally defined as the United States, Canada, Australia, New Zealand and the United Kingdom. We have added those the first time the word Anglosphere is used.

# Please clarify in the UK and Ireland are included with the Anglophone or European countries.

Ireland is analyzed as being part of the European category. We have added this to the text.

# Please list the five biggest non-Western countries in parenthesis the first time the phrase non-Western countries is used.

We have added the five biggest non-Western countries: Turkey, Japan, India, China, and Israel.

# In the third paragraph of the Discussion section, please reword the passage "It is important to note that gender assumptions are not black and white" to be more precise.# In the fourth paragraph of the Discussion section, please replace the first two sentences ("Our analyses […] statistical power") with one shorter sentence.

The abovementioned changes have been made.

9) FiguresTo help the presentation of the article, please do the following:# combine Figure 1, Figure 2 and Figure 3 into one figure with three panels.# combine Figure 6 and Figure 7 into one figure with two panels.# please add a colour scale to the present Figure 5.# please expand the caption for the present Figure 7 to better explain what has been measured and what is being predicted.

The abovementioned changes have been made.